# Navigating Real-World Randomized Clinical Trials: The ‘Parents as Teachers’ Experience

**DOI:** 10.3390/ijerph21081082

**Published:** 2024-08-16

**Authors:** Craig W. LeCroy, Carolyn Sullins

**Affiliations:** 1LeCroy & Milligan Associates, Inc., Tucson, AZ 85743, USA; 2Trewon Technologies, LLC, Leesburg, VA 20176, USA; csullins192@gmail.com

**Keywords:** randomized controlled trial, implementation research, community-based research, research challenges, parents as teachers

## Abstract

The Parents as Teachers Randomized Controlled Trial (PAT RCT) Case Study investigates the multifaceted impact of implementing the PAT RCT in Arizona, U.S.A., shedding light on both the positive and negative effects. There has been a recent focus on improving the implementation of RCTs in community settings, as this issue has not been fully addressed. This research presents a case study examining the implementation of a community-based RCT in home visitation. This study also addresses the strategies that can be employed to mitigate some of the challenges in the implementation of an RCT, offering valuable insights for future RCTs in the domain of home visiting. The PAT program, aimed at providing parent education and family engagement for children from birth to kindergarten, encompasses a range of services, including personal visits, group connections, child screenings, and community resource linkages. The Parents as Teachers Randomized Controlled Trial (PAT RCT) directly promotes health by educating parents about health and wellness as well as providing early child screenings and heath referrals, all of which enhance health outcomes through timely interventions and improved parental practices. Lessons from the study also aim to improve the implementation of future health-related RCTs, ensuring effective delivery and impactful results.

## 1. Introduction

The Parents as Teachers (PAT) evidence-based home visiting model (Parents as Teachers, 2023) has experienced significant growth and national attention. To evaluate its impact on families, a Randomized Control Trial (RCT) was commissioned by the PAT National Center (PATNC) with supporting funds from the Enterprise Holdings Foundation and Arizona First Things First. This article presents a comprehensive case study analysis of the PAT RCT, exploring its implications, challenges, and the broader context of implementing RCTs in community settings. The PAT model, focused on a developmentally centered parenting approach, aims to bolster families by fostering early childhood development and school readiness. It offers a suite of services, including personal visits, group connections, developmental and health screenings, and resource referrals. By equipping parents with knowledge and tools, the program aids in detecting developmental milestones, promoting school readiness, strengthening family bonds, and promoting child and family health and well-being [1].

### 1.1. The Significance of RCTs in Program Evaluation

RCTs are considered the gold standard in program evaluation due to their ability to provide a robust evidence base about an intervention’s effectiveness. By randomly assigning participants to treatment or control groups, RCTs eliminate many biases and offer compelling insights into causality. This standard is promoted by the Home Visiting Evidence of Effectiveness (HomVEE) review, which regularly evaluates the evidence of over 50 home visitation models. Very few models meet all the standards of evidence put forth by HomVEE. These rigorous standards include acceptable rates of overall and differential attrition, no reassignment of treatment and control, no confounding factors, no differences in data collection methods, and equivalent baseline variables or controls for these variables [2]. Despite the desirable qualities of an RCT, this rigor of study presents many complexities and challenges when implemented in a community setting. Some researchers have recently voiced concern about the compromised integrity in RCTs and some have called for a critical re-appraisal of RCTs [3].

Among the fifty-three home visiting models evaluated by HomVEE from 1979 to 2018, only twenty-four met the criteria for having at least one “high- or moderate-rated” study, including PAT. Specifically regarding PAT studies, one-hundred studies were identified, thirty were reviewed, and only two were rated high. This is only 6% of the total studies on PAT, which raises questions about why so few studies were able to meet the standards required to obtain a high rating. Furthermore, most of the studies were over 14 years old. Notably, HomVEE does not assess the impact of doing the studies themselves but rather focuses on the quality of the study methodology and results. This emphasis on accuracy is essential, but it does not account for considerations of propriety and feasibility in evaluating program effectiveness [4]. Given the expansion of home visitation services, understanding propriety and feasibility has become increasingly vital in evaluating the field’s effectiveness.

Researchers have written about the challenges in implementing RCTs in community settings, noting issues of ethical consideration [5,6], ensuring consistency in program implementation in real-world settings [7], problems with recruitment and retention [8], and adapting interventions to fit diverse community settings [9]. Ethical issues, which have been ignored, are being addressed, and authors have discussed the narrow scope and disruptive potential of RCTs, even pointing out how RCTs include potential dangers [10,11]. Although researchers are aware of these issues, there has not been much research that addresses these constraints [12]. In a review of RCT issues, Sibbald and Roland [13] noted that the rigorous control needed for an RCT should be balanced with the practical constraints the community settings impose. This study attempted to address these constraints by focusing on an early childhood RCT implementation. This provided an ideal opportunity to study how an RCT impacts a community and the implications of implementing RCTs to gather rigorous evidence [14].

To further this understanding, a case study was conducted on the Arizona PAT RCT that included a comprehensive review of the literature about the process of implementing RCTs and primary data collection from PAT RCT stakeholders to assess challenges. The case study method [15] is a research approach that involves in-depth exploration of a specific real-world phenomenon. The method involved an intensive investigation of a single case, the AZ PAT RCT implementation. The study used a variety of data collection methods including interviews, observations, document analysis, and surveys.

This study delves into the intricacies of conducting RCTs, shedding light on the challenges in meeting stringent research standards established by HomVEE and other national clearinghouses. It explores the difficulties associated with achieving these criteria and presents potential strategies for addressing them. Much like how the AZ PAT RCT aims to build evidence for the PAT model and advance the field of home visitation, the case study endeavors to provide valuable insights into the practicalities of conducting RCTs in real-world community settings. It presents not only the benefits but also the challenges faced during the implementation of the AZ PAT RCT, offering valuable guidance for future researchers embarking on similar endeavors.

### 1.2. Background of the Study

The PAT National Center (PATNC), headquartered in St. Louis, MO, USA, embarked upon undertaking a randomized controlled trial to continue to evaluate the effectiveness of the Parent as Teachers home visitation intervention. Their mission was to conduct a rigorous randomized controlled trial (RCT) of the Parents as Teachers (PAT) home visitation program, collaborating with four Arizona-based PAT affiliates. This ambitious endeavor aimed to add additional evidence to the PAT model’s status as an “evidence-based, home visiting service delivery model”, aligning with the standards set by the U.S. Department of Health and Human Services [3].

### 1.3. The AZ PAT RCT Study

The AZ PAT RCT study set out to compare a comprehensive range of child and caregiver outcomes over an 18-month period between two groups: those receiving PAT services (Treatment Group), *N* = 456, and those not receiving PAT services (Control Group), *N* = 231. The Treatment Group consisted of families enrolled in four distinguished Blue Ribbon PAT program sites in Arizona.

Upon obtaining informed consent and random assignment to study groups, all participating families underwent assessments conducted by the evaluation team at four key time points: enrollment, 6 months, 12 months, and 18 months post-enrollment. During the study, families engaged with child development specialists, who collected parent- and child-level data at six-month intervals, providing incentives and age-appropriate books to families after each visit. Furthermore, the Treatment Group received PAT services from a Blue Ribbon certified affiliate, including personal visits with a Parent Educator. The case study explores the broader implications of conducting the AZ PAT RCT, encompassing its effects on various stakeholders, both positive and negative, while also highlighting successful strategies to mitigate adverse consequences.

The study’s central focus lies in exploring the experiences of PAT program stakeholders, providers, and participants involved in the RCT study. The following outlines the overarching evaluation questions that guide this investigation:What has been the impact of participating in and helping conduct an RCT in the field?What are the perceived and demonstrated benefits of conducting an RCT?What motivated people to take part in the AZ PAT RCT study?What have been the most challenging aspects of the AZ PAT RCT study?What has worked well to mitigate these challenges?How well do stakeholders understand and accept the AZ PAT RCT?What can be recommended in terms of mitigating the impacts of RCT’s on communities?

## 2. Materials and Methods

The AZ PAT RCT Case Study employs a comprehensive case study methodology to delve into the intricacies of conducting an RCT in community-based settings. The case study approach [16] is particularly apt for gaining a nuanced understanding of complex phenomena, aligning with the intensive exploration of the AZ PAT RCT’s implementation in Arizona.

To collect data, mixed methods were employed, encompassing online surveys and semi-structured interviews, drawing inspiration from prior case studies in the field of home visiting programs [17]. These data collection methods involved a broad spectrum of perspectives, including local and national PAT program leaders and associates, AZ PAT RCT Parent Educators, RCT evaluators, data collectors, and parent participants. An iterative approach to study design allowed for findings from initial stakeholder data collection to inform subsequent data collection methods and questions.

### 2.1. Data Collection Methods

The study employs a mixed-methods approach encompassing two main data collection methods: individual and dyadic interviews with stakeholders and a survey of PAT Parent Educators.

Interviews: A semi-structured interview protocol was developed in alignment with the overarching evaluation questions. Interviews were conducted using the Zoom platform with the informed consent of participants, and a dedicated notetaker was either present during the interviews or reviewed recorded video sessions to compile detailed notes. Telephone interviews with parents were conducted, and notes were taken during these conversations. Interviews were conducted with various stakeholders, including the following:

AZ PAT RCT data collectors (*N* = 2)

Evaluators (*N* = 2)

Evaluation administrators (*N* = 2)

PATNC leadership (*N* = 4)

First Things First (AZ PAT RCT funder/“best practice” expert) (*N* = 1)

Parent Partners Plus Central Intake staff (*N* = 1)

PAT administrators at AzCA, CFR, CC, and SUSD (*N* = 5)

AZ PAT RCT parent participants (*N* = 4)

Evaluators of other home visiting programs (*N* = 2)

In most cases, interviews were conducted individually. However, because some administrators worked together as a team of two, they opted to be interviewed as a dyad.

Survey of Parent Educators: An online survey was designed and administered to elicit perspectives from PAT Parent Educators who had served families in the Treatment Group of the RCT study. The survey included questions about the participants’ roles in PAT and how long they had been involved in them. The main questions were Likert-scale items about how they saw the benefits and drawbacks conducting the PAT study for themselves and the families whom they served.

All 38 Parent Educators from the four study sites were invited to participate in the survey, and a commendable 82% response rate was achieved, with 31 educators completing the survey.

### 2.2. Analyses

The transcripts of the interviews were assessed using N*Vivo 14 (Lumivero, Denver, CO, USA) software to search for emergent themes using an iterative process. The resulting codes were checked for replicability by having two coders discuss and compare themes. The results of the quantitative portions of the survey were analyzed using descriptive statistics (counts, percentages).

## 3. Results

### 3.1. Interviewee Perspectives on RCT Benefits

Insights gathered from interviews with PATNC and Arizona affiliate representatives reveal several interrelated benefits of successfully implementing an RCT, including the following:Building the evidence base for PAT.Enhancing program credibility for funders and potential implementers.Increasing the likelihood of additional funding to improve or expand program implementation.

Interviews with other program staff and parent participants further highlight the personal, professional, and societal advantages of participating in the AZ PAT RCT. Interviewees have gained a deeper understanding of the program’s implementation and appreciate the RCT’s widespread recognition among PAT affiliates, leadership, and stakeholders. This emphasis on the value of research findings enables programs like the PAT program to receive favorable ratings from clearinghouses, enhancing their eligibility for various funding sources.

Table 1 displays some quotes from the various PAT stakeholders regarding what they viewed as the benefits of RCT. 

### 3.2. Parent Educator Perspectives on RCT Benefits

The survey of Parent Educators sought to gauge their opinions on the benefits of RCTs, and the results were generally positive. A significant proportion of respondents expressed agreement or strong agreement with statements related to RCT benefits. Some key findings from the survey of Parent Educators (*N* = 31) are as follows.

Over 90% agreed or strongly agreed that “There are useful benefits to a home visiting program being involved in an RCT study”.Similarly, over 90% agreed or strongly agreed that “The results of RCTs help home visiting programs improve in the long run”.The importance of research demonstrating the effectiveness of the PAT program was acknowledged as “very important” by 77% and “important” by 19% of Parent Educators surveyed.Regarding the importance of PAT being ‘evidence-based’ to the parents they work with, 48% said it was “very important”, and 39% said it was “important”.Furthermore, 71% of Parent Educators agreed or strongly agreed that “the benefits of an RCT study outweigh the burdens”.

Figure 1 displays additional details about how the Parent Educators viewed the benefits of the RCT study.

These findings indicate that Parent Educators who work directly with families participating in the AZ PAT RCT recognize the value and benefits of conducting an RCT.

### 3.3. Challenges and Complexities of RCTs

While RCTs are highly regarded for their ability to evaluate intervention effectiveness rigorously, there are several challenges and concerns associated with their implementation in community settings. Both the surveys and the qualitative themes from the interviews uncovered the following issues:Ethical Concerns: Mistrust and ethical concerns arise due to the random assignment of families to control groups, potentially denying some families access to services. A significant portion (60%) of Parent Educators surveyed believed that RCTs are unfair to those placed in the control group.Recruitment Challenges: RCTs necessitate a higher recruitment goal and pose difficulties in recruiting each family, as twice as many participants are required to ensure equal randomization. Parent Educators expressed concerns that RCTs make it harder to enroll enough families into home visitation programs; of all the challenges listed, this was the item with which there was the strongest agreement.Participant Confusion: Some parents may not fully understand the difference between the research data collection process and the intervention itself, leading to confusion when data collectors are involved in home visits.Challenges in Measuring Outcomes: Selecting outcomes to measure the impact of home visitation programs can be challenging due to the complex and multifaceted nature of the interventions and the broader environmental factors that affect families.External Factors: Contextual factors, such as the COVID-19 pandemic, can significantly impact program implementation and data collection for RCTs, leading to additional challenges.

Early in the implementation of the RCT, the PAT staff and the evaluators recognized that the RCT posed several challenges to the PAT providers and the families whom they served. Therefore, the evaluation employed numerous strategies to ameliorate the challenges while maintaining a robust design.

### 3.4. Strategies to Mitigate RCT Challenges

Figure 2 presents details on what the sample of PAT staff found most challenging about the RCT, as per our survey.

To address the challenges associated with RCTs, several strategies were employed in the AZ PAT RCT:Adjusting Randomization Ratio: To mitigate concerns about reduced program enrollment numbers, the randomization ratio was adjusted to 2:1 (PAT treatment group to control group), which helped maintain funding levels for program sites.Providing Limited Study Bypasses: The study introduced bypasses where agencies were given a set number of “opt outs” for the study, they were withdrawn from the study and received PAT services directly, addressing worker concerns about families not receiving services when needed.Offering Resources to Control Group: Control group families were provided with information on community resources similar to PAT services, reducing staff concerns about families not having access to services.Educating Stakeholders: Establishing trust and open communication with program staff and participants was crucial in encouraging cooperation with RCT requirements. Communication emphasized the benefits for both control and treatment groups.Incorporating Qualitative Methods: Qualitative methods and lived experience experts provided personal stories and perspectives that helped participants feel included and validated.Highlighting Benefits: Materials describing the AZ PAT RCT emphasized the benefits for both groups, aiming to make research participation empowering and beneficial for all participants.

These strategies aimed to mitigate recruitment challenges, ethical concerns, participant confusion, and other issues related to conducting RCTs in a community context.

### 3.5. Balancing Program Funding and Evidence

Balancing the need for robust evidence through RCTs with the necessity for timely program funding decisions is a complex challenge. Funders often require evidence of program effectiveness, but RCTs can be time-consuming and may delay decision-making. Striking a balance between obtaining rigorous evidence and ensuring ongoing program funding is essential for sustaining effective interventions.

Challenges in measuring outcomes, particularly in the context of multifaceted interventions like home visitation programs, require careful consideration of measurement tools and methodologies. The validated measures were thoughtfully selected for the AZ PAT RCT to increase the likelihood of accurately assessing outcomes.

The impact of external factors, such as the COVID-19 pandemic, on program implementation and data collection underscores the need for flexibility and adaptability in conducting RCTs. These external challenges can significantly affect recruitment, data collection, and program delivery, necessitating adjustments to study protocols.

In summary, while RCTs are considered the gold standard for program evaluation, their implementation in community settings presents unique challenges related to ethics, recruitment, participant understanding, measurement, and external factors. Employing strategies to address these challenges and striking a balance between obtaining rigorous evidence and ensuring program sustainability are essential for conducting successful RCTs in the context of home visitation programs like PAT.

### 3.6. Alternative Evaluation Designs

Concerns over RCTs have highlighted alternative study designs. This is one of the most complete ways of mitigating RCT challenges, but not without some constraints and limitations. Although not as rigorous, alternative designs can excel in practicality and feasibility. It is noteworthy that the American Evaluation Association wrote a policy statement that attempted to address misunderstandings about the perceived exclusive value of RCTs. The AEA noted “RCTs are not the only studies capable of generating understanding of causality; RCTs can be misleading; they may not be sensitive to local culture; and RCTs are sometimes not useful due to ethical concerns”. However, this sentiment is not widely acknowledged among policy makers or researchers. More specifically, quasi-experimental designs and propensity score matching are two approaches that offer potential solutions in situations where RCTs face limitations [18,19]. Quasi-experimental designs involve comparing groups that are naturally exposed to the intervention or not, rather than randomly assigning participants to study groups. These designs can be useful when randomization is impractical or deemed unethical. Propensity score matching is a statistical technique used to match individuals or groups based on their propensity or likelihood of receiving a specific intervention. This method aims to create comparable groups by accounting for potential confounding variables that may influence treatment assignment and outcomes. Propensity scores are estimated using observed covariates and matched individuals are then compared to evaluate the intervention’s impact [18]. Analysis of outcome findings only for intervention group participants who go on to receive services might also be explored. While impacting the intent-to-treat and the assumptions of the model, these results could also potentially speak more accurately to the benefits participants in the program itself receive [19]. Perhaps both analyses might be considered. Alternative designs and the benefits and downsides of their utilization should continue to be explored as possible alternatives to building an evidence base.

## 4. Discussion

A sound evaluation is always a balancing act among the standards of propriety, feasibility, utility, and accuracy [4]. According to Daniel Stufflebeam (personal communication), accuracy is often considered the primary standard, but it cannot be achieved without propriety, feasibility, and utility. When it comes to accurate data regarding causality, the RCT has been and continues to be the gold standard of research and evaluation designs. However, RCT’s are very challenging to implement with propriety and feasibility, especially in the context of home visiting programs. These challenges are not widely discussed in research articles.

The biggest concern faced in this study was the denial of treatment to families who were assigned to the control group. This issue has been raised by other researchers who have serious concerns about how RCTs disrupt communities. In a review of these issues, Picciotto discusses how many established evaluators have transitioned from ridged adoption of RCTs to answer important policy questions to a much more moderate approach. For example, Cronbach began to question the external validity of RCTs, and Campbell moved from advocating only experimental methods to seeing value in expert qualitative judgment [20]. Further, the contexts in which an RCT is implemented often hamper the design’s utility and feasibility. Table 2 summarizes the potential barriers to RCTs, their dilemmas to evaluation standards, and how the AZ PAT RCT adapted to these dilemmas.

This case study, through an in-depth case study, provides a comprehensive picture of RCTs and their benefits and limitations. The findings highlight the complexity of the study design and the implications at the individual participant, program, community, and societal levels of the selection that is ultimately made.

## 5. Conclusions

Program evaluations, such as the AZ PAT RCT, aim to meet standards of propriety, feasibility, utility, and accuracy [21]. These standards need to be brought back into the picture when conducting community-based RCTs [21]. However, achieving certain standards often necessitates trade-offs. The findings from the literature review and data collected in this study elucidate the importance of this study and the benefits and challenges of conducting RCTs and outline strategies to address these challenges. These findings are consistent with the field of evaluation’s focus on utilization [21].

The AZ PAT RCT Case Study contributes significantly to our understanding of conducting RCTs in the realm of home visiting. It illuminates the challenges, strategies, and recommendations for researchers aiming to navigate the complex landscape of community-based trials, ultimately advancing the field of home visitation and evidence-based practices.

There are a number of limitations to this study. It is important to note that this study does not constitute a meta-evaluation of the AZ PAT RCT’s implementation in Arizona. Furthermore, there was a limited sample of parents upon which the study is based. Additionally, the study does not incorporate feedback from parents who declined or dropped out of the RCT due to difficulties in reaching this particular population. Therefore, the findings predominantly reflect the perspectives and experiences of staff and program stakeholders.

This case study aligns with current critiques of the complexity of study design decisions and their implications at various levels [11]. It underscores the importance of adapting evaluation methods to the specific context and challenges of the program being assessed while striving to meet the highest standards of research and evaluation.

## 6. Clinical Implications

The Parents as Teachers Randomized Controlled Trial (AZ PAT RCT) Case Study in Arizona highlights several clinical implications for implementing RCTs in community-based early childhood home visitation programs. Key takeaways include the need for ethical considerations, particularly in treatment allocation to control groups, and the importance of clear participant communication to avoid confusion. The study emphasizes the necessity of adaptable and responsive program designs that respect local cultural contexts and practical constraints. It also underscores the importance of selecting appropriate outcome measures and balancing rigorous research standards with the realities of community settings. Additionally, the case study suggests exploring alternative evaluation designs, like quasi-experimental studies, for more practical and ethical research approaches. This study provides valuable insights for clinicians, researchers, and program administrators, stressing the importance of flexibility, ethical conduct, and community engagement in RCT design and implementation to ensure effective and community-responsive interventions.

## Figures and Tables

**Figure 1 ijerph-21-01082-f001:**
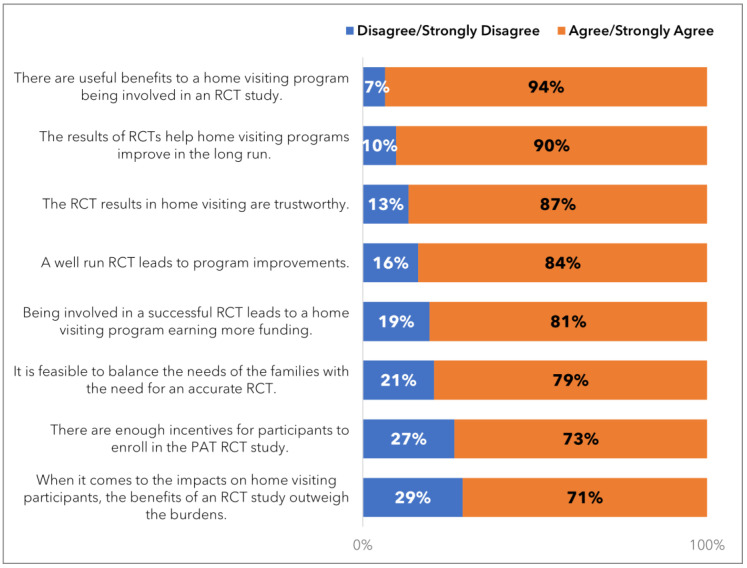
Parent Educators’ opinions on the benefits of RCTs.

**Figure 2 ijerph-21-01082-f002:**
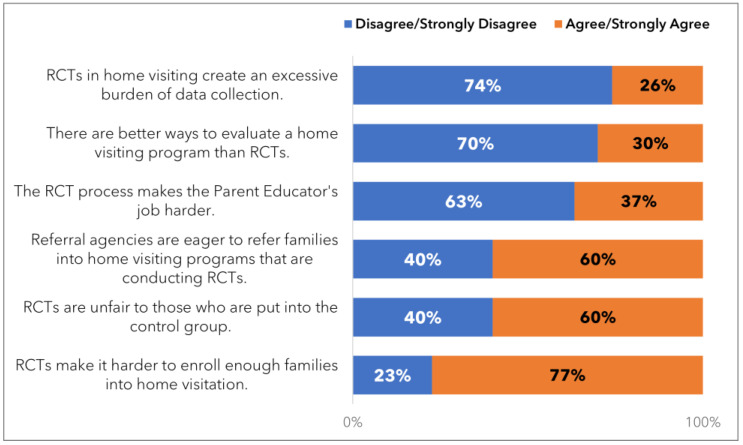
Challenges in conducting an RCT.

**Table 1 ijerph-21-01082-t001:** Benefits of an RCT according to Stakeholders.

Benefits of the AZ PAT RCT Study to Stakeholders	Stakeholder Quotes
Building the evidence base for PAT	- “Conducting RCTs resulted in getting onto various clearinghouses, which enabled in securing federal funding, such as SAMHSA’s clearing house, HomVee, and Family’s First Prevention Services act clearing house.” (PATNC) - “The idea that early childhood education being studied was exciting. It needs a deeper understanding of how it works, this study was big and had the potential to get national attention.” (Site Leader)
Credibility among potential benefactors or subsequent implementers	- “Seeing the benefits the RCT could bring to our own program and the understanding of how PAT influences families. The reason why we embarked on this study is to be able to have those benefits and be able to reflect those throughout, not just the state but the nation.” (Site Leader) - “I think there’s some new opinions about evaluations and community-based evaluations, but so many that’s seen as the gold standard, and we always want to be able to show the impact and effectiveness of Parents as Teachers when we’re dealing with funders and or implementers, and especially from the point of view with decision-makers legislators.” (National Office)
Securing additional funding	- “Many decisions are made by funders that are data-driven. It’s not just showing them a family and having them do their testament because sometimes that’s just one family. Maybe it will open doors for more funding, not just in Arizona but in other places. Being able to show and demonstrate with more updated information and data will be great for us, not just for our current programs but also for future funding.” (Site Leader) - “It is hard to get funding from particular streams if you have evidence that is not [from] an RCT.” (National Office)
Additional benefits to self and others	- “Being in the study helped me stay on top of program data.” (Site Leader) - “Really seeing how families that are being served and enrolled in the study are weathering the storm of COVID-19 a little bit easier.” (National Office) - “I like being in studies, you can give a voice to people.” (Treatment Group Parent) - “The questions helped me reflect more as myself as a mom, it made me feel like I was doing well.” (Control Group Parent)

**Table 2 ijerph-21-01082-t002:** Evaluation standards, potential barriers to RCTs, and AZ PAT RCT mitigation approaches.

Evaluation Standard	Potential Barrier to RCTs	PAT RCT Mitigation Approaches
Propriety	Parents’ concerns about being “used” as a research participant	Provided detailed informed consent, offered parents incentives, established exclusion criteria for families in most need.
	Parents’ concerns about being denied services as members of the “control group”	Control group was provided resources to address requested needs.
	The neediest parents could be denied home visiting services	Home visiting set-asides for the neediest families (who were not included in the study)
Feasibility	Too expensive and too difficult to enroll enough families into both the treatment and control groups	Enrolled two treatment participants for every control group participant
Accuracy	Differing rates of attrition among treatment and control groups	Will need to evaluate attrition rates with control and treatment
	Control group could receive similar services elsewhere	All participants—treatment and control—are asked about various other services to control in the analysis.
	Intent-to-treat diluting findings	Explore conducting analyses for all intervention participants (standard intent-to-treat) and only for participants engaging in services to see if findings differ.
Utility	Results are needed in a timeframe that a funder needs	Can be an issue when funders need more timely data than RCT’s can feasibly deliver
Utility/Feasibility	Unexpected, external factors can lead to difficulties running the program, administering the study, and/or interpreting the results.	The COVID-19 pandemic created profound stressors to families, and necessitated shifts from in-home to virtual visits from PAT Parent Educators as well as data collectors. The RCT evaluation not only took these changes into account, but initiated a sub-study of (1) the adaptations to the PAT implementation and (2) the impact of the pandemic on the PAT treatment recipients versus the control group

## Data Availability

Data collected for this project is available from the authors.

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
