# Peer review of "Navigating Real-World Randomized Clinical Trials: The ‘Parents as Teachers’ Experience"

_ijerph, 2024, doi:10.3390/ijerph21081082_

Round 1

Reviewer 1 Report

Comments and Suggestions for Authors

I enjoyed reading the manuscript and think it is an important contribution to the literature.  I only have a few suggestions:

1. You did not mention the reliability process regarding the qualitative coding. That information could be added. 

2. It is not clear how many families participated as recipients of the services on the two groups.  That information is important.

3.  A practice that was not clearly mentioned in the article is to offer the intervention to the participants in the control group after the data collection has been completed.  This practice allows all research participants to benefit from the intervention, including those in the control group.  This should be included and discussed in the article. 

Author Response

  1. YComment: you did not mention the reliability process regarding the qualitative coding. That information could be added.
  2. Response: Added comment about 2 coders discussing and comparing themes for reliability.
  3. It is not clear how many families participated as recipients of the services on the two groups.  That information is important.
  4. Response: added numbers, 456 in treatment group, 231 in control group.
  5.  A practice that was not clearly mentioned in the article is to offer the intervention to the participants in the control group after the data collection has been completed.  This practice allows all research participants to benefit from the intervention, including those in the control group.  This should be included and discussed in the article. 
  6. Response: Participants were not offered services after study completion. This was due to the length of the study, access to other services by control group, and the desire to conduct long term follow up.

Reviewer 2 Report

Comments and Suggestions for Authors

This paper deals with important issue of investigation the multifaceted impact of implementing the Parents as Teachers Randomized Controlled Trial in Arizona, U.S.A. The study examines both positive and negative effects of this community-based RCT in home visitation, offers strategies to mitigate implementation challenges, and provides valuable insights for future RCTs in the domain of home visiting. Authors states that PAT program aims to provide parent education and family engagement services, promoting health through education, early child screenings, and health referrals, thereby enhancing health outcomes and improving parental practices.

Study is based on case study methodology with aim to explore the complexities of conducting a randomized controlled trial (RCT) in community-based settings in Arizona. It successfully employs mixed methods for data collection, including online surveys and semi-structured interviews, gathering perspectives from various stakeholders. Authors states that iterative approach allowed initial findings to inform subsequent data collection methods and questions. The primary data collection methods seem to be adequate related to the aim of the research.

Manuscript is written in clear manner and well structured. Results are presented in comprehensive and approachable way that can be useful for professionals in educational theory and practice.  However, discussion section should be more elaborate and should contain more comparisons with recent and relevant findings of similar studies and possibly comparisons with more examples from direct practice.    Conclusions are clear and coherent with presented results and discussion.

In general, it is well written and structured paper with valid and important findings that can be improved by providing more and critical insights based on comparison with similar studies in discussion section.

Author Response

Reviewer Comment: However, discussion section should be more elaborate and should contain more comparisons with recent and relevant findings of similar studies and possibly comparisons with more examples from direct practice. 

Response: Discussion section has been rewritten and includes more comparisons.

Reviewer 3 Report

Comments and Suggestions for Authors

The abstract and introduction require further references on the appropriateness of RCTs.

Ethical considerations require explanation and depth, citing relevant references.

Analysis requires further depth reflecting further literature on relevance of RCT's.

Further analysis in the conclusions required in the discussion on methodology and importance of this study.

Author Response

Reviewer comment: The abstract and introduction require further references on the appropriateness of RCTs.

Response: References were added as suggested.

Reviewer comment: Ethical considerations require explanation and depth, citing relevant references.

Response: Ethics discussion expanded with new references.

Reviewer: Analysis requires further depth reflecting further literature on relevance of RCT's.

Response: New discussion addresses this and includes further literature.

Reviewer: Further analysis in the conclusions required in the discussion on methodology and importance of this study.

Response: Conclusions address the importance of the study